# Biomechanical Analysis of Axial Gradient Porous Dental Implants: A Finite Element Analysis

**DOI:** 10.3390/jfb14120557

**Published:** 2023-11-23

**Authors:** Chunyu Zhang, Yuehong Wang

**Affiliations:** 1Xiangya Stomatological Hospital, Central South University, No. 72 Xiangya Street, Kaifu District, Changsha 410008, China; zhangchvnyu@csu.edu.cn; 2Xiangya School of Stomatology, Central South University, No. 72 Xiangya Street, Kaifu District, Changsha 410008, China; 3Hunan 3D Printing Engineering Research Center of Oral Care, No. 64 Xiangya Street, Kaifu District, Changsha 410008, China

**Keywords:** gradient porous implants, finite element analysis, biomechanics, elasticity modulus

## Abstract

The porous structure can reduce the elastic modulus of a dental implant and better approximate the elastic characteristics of the material to the alveolar bone. Therefore, it has the potential to alleviate bone stress shielding around the implant. However, natural bone is heterogeneous, and, thus, introducing a porous structure may produce pathological bone stress. Herein, we designed a porous implant with axial gradient variation in porosity to alleviate stress shielding in the cancellous bone while controlling the peak stress value in the cortical bone margin region. The biomechanical distribution characteristics of axial gradient porous implants were studied using a finite element method. The analysis showed that a porous implant with an axial gradient variation in porosity ranging from 55% to 75% was the best structure. Under vertical and oblique loads, the proportion of the area with a stress value within the optimal stress interval at the bone–implant interface (BII) was 40.34% and 34.57%, respectively, which was 99% and 65% higher compared with that of the non-porous implant in the control group. Moreover, the maximum equivalent stress value in the implant with this pore parameter was 64.4 MPa, which was less than 1/7 of its theoretical yield strength. Axial gradient porous implants meet the strength requirements for bone implant applications. They can alleviate stress shielding in cancellous bone without increasing the stress concentration in the cortical bone margin, thereby optimizing the stress distribution pattern at the BII.

## 1. Introduction

In the clinical application of dental implants, intraosseous stress must be maintained within a certain range to ensure physiological balance. Excessive stress can cause bone resorption or fatigue destruction of the implant, and too little stress may lead to disuse atrophy of bone, resulting in bone defects [1,2,3,4]. According to the requirements of implant biomechanical compatibility, the more uniform and dispersed the intraosseous stress distribution is, the more conducive it is to promote the reconstruction of bone tissue around the implant. In contrast, if the stress distribution is too concentrated, the coordination between the implant and the surrounding bone becomes poor, which increases the risk of bone resorption [5,6].

Titanium and its alloys have a high strength coefficient, ideal biocompatibility, and excellent corrosion resistance [7] and can obtain excellent bioadhesion after surface treatment [8,9], making them the preferred material for dental implants. However, the elastic modulus of titanium and its alloys (110 gigapascals (GPa)) is significantly different from that of the bone tissue (17–20 GPa for cortical bone and 1–4 GPa for cancellous bone). The mismatch of the elastic modulus between implants and bone tissue can trigger the stress shielding of the trabecular bone and progressive trabecular osteoporosis [10,11,12]. Therefore, the elastic modulus of implant materials is a key factor affecting the biomechanical compatibility of implants [13,14].

A porous implant with high porosity exhibits a low elastic modulus, which is an ideal solution for solving the mismatch between the implant and alveolar bone [15,16,17,18,19]. However, a previous study found that homogeneous porous implants have limited practical applications [20,21]. Firstly, the strength of homogeneous porous implants decreases significantly with the increase in porosity. Specifically, the peak stress in the implants even exceeds the yield strength of pure grade IV titanium in high-porosity implants with over 65% porosity. Secondly, the porous structure not only promotes stress transfer and alleviates stress shielding in cancellous bone but also significantly increases the peak stress value in the cortical bone margin, which aggravates the resorption of the alveolar process. Finally, the homogeneous porous structure cannot achieve an ideal stress distribution at the bone–implant interface (BII) as a whole due to the structural heterogeneity of the alveolar bone in the axial direction, perhaps because of the numerous stress-shielding areas in the cancellous bone region corresponding to the root 1/2 of the implant.

Herein, we designed a high-strength implant structure and constructed an axial gradient porous (GP) implant with low porosity in the crown direction region and high porosity in the root direction region to ensure the strength of the porous implant itself, control the peak stress value in the cortical bone margin area, and reduce the stress shielding at the BII.

## 2. Materials and Methods

### 2.1. Construction of a Three-Dimensional (3D) Geometric Model

#### 2.1.1. Construction of the Mandibular Model

A 3D model of the mandible was constructed using UG NX 10.0 software, simplifying and simulating the real mandibular morphology, with a length of 14.4 mm in the mesial and distal directions and 15.5 mm in the buccal and lingual directions and a height of 27 mm. The thickness of the outer cortical bone was 1.3 mm, and the inner cancellous bone was surrounded by the cortical bone in the buccal, lingual, upper, and lower directions (Figure 1a) [22].

#### 2.1.2. Construction of a Standard Implant Model for the Control Group

The standard implant model for the control group was constructed using UG NX 10.0 software. Since the connection between the implant and abutment was an irrelevant variable in this study, to follow the principle of appropriate simplification in finite element modeling, an integrated implant and abutment model was constructed. The main component of the standard implant model was a cylindrical-like structure with a height of 10 mm and a diameter of 4 mm. The total height of the abutment was 7.4 mm, and it was connected to the top of the implant in an arc. The diameter of the round surface connecting the abutment to the restored crown’s gingival margin part was 5.4 mm, and the diameter of the upper surface was 3 mm (Figure 1b).

#### 2.1.3. Construction of homogeneous porous (HP) implant models

HP implants with a pore size of 500 μm and porosity of 35% (T1), 45% (T2), 55% (T3), 65% (T4), and 75% (T5) were constructed [23,24] and used to analyze the effect of porosity on the stress distribution of BII. The porous structure design is shown in Figure 1c [25,26]. The theoretical formula of porosity is shown in Formula (1) [27].
(1)$$\;Porosity\left(p\right)=1-\frac{V_{HP\;implant}}{V_{control\;group}}$$
where *V* indicates the measured volume of the implant model. HP implant models are shown in Figure A1.

#### 2.1.4. Construction of High-Strength Designed Implant Models

For the HP implants, the T3 model showed a relatively balanced stress distribution pattern. Therefore, based on the T3 model, five types of porous implants with a strength enhancement design were constructed as follows: (1) T3P: a solid pillar with a diameter of 2.4 mm was added to the center of the implant; (2) T3C: the implant neck was solid, and only the porous structure was added in the corresponding area of the cancellous bone; (3) T3B: the neck and apex of the implant were solid, and only the porous structure was added in the corresponding area of the cancellous bone, except for the apical region; (4) T3PC: a solid pillar in the center of the implant was combined with a solid neck design; and (5) T3PB: a solid pillar in the center of the implant was combined with a solid design at the neck and apex.

#### 2.1.5. Construction of the Axial GP Implant Model

Based on the finite element analysis (FEA) results of high-strength designed implant models, GP implant models with low porosity in the crown direction region and high porosity in the root direction region were constructed. Ten sets of models were established, namely T1/2, T1/3, T1/4, T1/5, T2/3, T2/4, T2/5, T3/4, T3/5, and T4/5 (Table 1). GP implant models are shown in Figure A2.

#### 2.1.6. Assembly of FEA Models

The implant model of each pore parameter and the mandible bone segment model were assembled in UG NX 10.0 software and exported to the stp format file.

### 2.2. Pre-Processing of FEA

Meshing was performed using Altair HyperMesh software. A 0.4 mm second-order C3D10M grid was the mainstay, and a 0.2 mm C3D10M grid was used at the BII for local encryption. After meshing, the quality of grid cells was checked to ensure that the overall grid quality met the standard. The details of the finite element mesh are shown in Figure 1d. The number of elements and nodes for each model are shown in Table 2. The material mechanical parameters were assigned to the model components (Table 3).

Assuming that the models used a continuous, homogeneous, and isotropic linear elastic material, the implant and alveolar bone formed complete osseointegration, and there was no tangential relative displacement at BII. Therefore, the contact between the implant and the alveolar bone was set as a tie constraint. A fixed boundary constraint was created on the mesial and distal planes of the mandible bone segment model [28], with U1 = U2 = U3 = UR1 = UR2 = UR3 = 0.

The stress on the implants and surrounding bone tissue was simulated under average and maximum occlusal forces, respectively. The average occlusal force was achieved via the application of an axial force of 120 N, whereas the resultant force of 118.2 N at 15° with the long axis of the implant simulated the maximum occlusal force, which was synthesized by three forces: 114.6 N axially, 17.1 N buccolingually, and 23.4 N mesiodistally [29].

**Table 3 jfb-14-00557-t003:** Mechanical parameters of structural materials [30,31].

Materials	Young’s Modulus/MPa	Poisson’s Ratio
Cortical bone	13,700	0.30
Cancellous bone	1370	0.30
Implant	110,000	0.35

### 2.3. Analytical Methods and Indicators

The stress distribution was numerically simulated using the Abaqus 6.14 FEA software package. The initial increment size was set to 0.01. Analytical objects included all elements and nodes in the BII. Analysis indicators were as follows: (1) the change in the equivalent stress value from the neck to the apex on the BII longitudinal reference line, (2) the percentage of nodes with an equivalent stress value at the BII within the interval of S < 1.6 MPa, 1.6 MPa ≤ S < 40 MPa, and S ≥ 40 MPa, and (3) the peak value of equivalent stress.

## 3. Results

### 3.1. Stress Distribution of HP Implants

#### 3.1.1. Axial Variation Trend of Equivalent Stress

The stress nephograms of T0–T5 models under a vertical load of 120 N and an oblique load of 118.2 N are shown in Figure 2, and the axial variation trend of equivalent stress values is presented in Figure 3. The results were as follows: (1) Nodes with S ≥ 40 MPa were mainly distributed in the alveolar ridge margin, while nodes with S < 1.6 MPa were mainly distributed in the cancellous bone region. (2) Fault stress occurred at the junction of the cortical bone and cancellous bone. (3) The peak value of equivalent stress at the cortical bone margin (X = 0 mm) increased with an increase in the porosity of HP implants.

#### 3.1.2. Interval Distribution of Equivalent Stress

The results of the interval distribution of equivalent stress are shown in Table 4. The area of S < 1.6 MPa at the BII decreased gradually with an increase in porosity, while the area of 1.6 MPa ≤ S < 40 MPa and S ≥ 40 MPa increased gradually. From the absolute value perspective, the proportion of nodes whose stress value was within 1.6 MPa ≤ S < 40 MPa increased significantly, whereas the proportion of nodes whose S ≥ 40 MPa increased significantly from the growth rate perspective (Figure 4). The growth rate is equal to $1+\left(T_{i}-T_{0}\right)/T_{0}$, *i* = 1, 2, 3, 4, 5. It should be noted that, under a 120 N vertical load, the number of nodes whose S ≥ 40 MPa in the T0 model is zero (i.e., the denominator is zero). Since the growth rate cannot be calculated in this case, it is ignored in Figure 4d.

#### 3.1.3. Peak Equivalent Stress

Peak values of equivalent stress in cortical bone in T0–T5 models are shown in Figure 5a. Regardless of an oblique load of 118.2 N or a vertical load of 120 N, the maximum equivalent stress values in the cortical bone of HP implants with each porosity far exceeded the upper limit of the ideal bone stress value (40 MPa). Under an oblique load of 118.2 N, the maximum equivalent stress values in T4 and T5 were 787.9 MPa and 1128 MPa, respectively (Figure 5b), which far exceeded the yield strength of pure grade IV titanium of 483 MPa.

After a comprehensive analysis of the aforementioned data, we found that the area of 1.6 MPa ≤ S < 40 MPa in the T3 model was only slightly lower than that of T4 and T5; the area of S ≥ 40 MPa was significantly lower than that of T4 and T5 and was the same as that of T1 and T2. Meanwhile, the maximum equivalent stress value in T3 was far below 483 MPa. Therefore, we chose the T3 model as the basis for follow-up research.

### 3.2. Stress Distribution of High-Strength Designed Implants

#### 3.2.1. Peak Equivalent Stress in Implants

Under a 120 N vertical and 118.2 N oblique load, peak equivalent stress values of T3P, T3C, T3B, T3PC, and T3PB were all less than that of the T3 model. Among them, peak equivalent stress values of T3PC and T3PB decreased significantly and were similar to those of T0. Therefore, compared with HP implants, the five strength-enhancing designs in this study improved the mechanical strength of the implants. Figure 6a shows the peak equivalent stress value in a porous implant with each high-strength design.

#### 3.2.2. Interval Distribution of Equivalent Stress

The stress nephograms of the five high-strength designed implants are shown in Figure 6b, and the interval distribution of equivalent stress is presented in Table 5 and Figure 6c. The results showed that under a 118.2 N oblique and 120 N vertical load, the distribution area of the T3PC model within the optimal stress interval of 1.6 MPa ≤ S < 40 MPa was 32.67% and 28.39%, respectively, second only to the T3P model. Meanwhile, under a 118.2 N oblique load, the distribution area of S ≥ 40 MPa in the T3PC model was 0.18%, which was only 1/3 of T3P, and the distribution area of S ≥ 40 MPa in the T3PC model was 0% under a vertical load of 120 N. Therefore, the equivalent stress distribution of the T3PC model was the best at the BII among the five high-strength designed porous implants.

### 3.3. Stress Distribution of GP Implants

#### 3.3.1. Peak Equivalent Stress

Based on the T3PC model, a strength enhancement design was added to GP implants to ensure that the peak equivalent stress value in these implants would not exceed the yield strength of pure grade IV titanium. Peak equivalent stress values in the ten groups of GP implants are shown in Figure 7a. The results showed that all GP implants met the strength requirements for implant applications.

Peak equivalent stress values of cortical bone in GP implants at the BII are shown in Figure 7b and Table 6. Compared with HP implants, GP implants had significantly lower and more stable maximum equivalent stress values, whether under a 118.2 N oblique or 120 N vertical load, with higher safety.

#### 3.3.2. Axial Variation Trend of Equivalent Stress

The axial variation trend of equivalent stress values of GP implants under a 120 N vertical and 118.2 N oblique load is shown in Figure 8. The results were as follows: (1) under a 118.2 N oblique load, the Y-value corresponding to X = 0 was only slightly greater than 40 MPa and not more than 50 MPa in each GP implant. (2) Under a 120 N vertical load, the Y-value corresponding to X = 0 was equal to about 20 MPa, which is consistent with the values presented in Figure 7b. (3) The X-value corresponding to Y(X) = 1.6 MPa gradually shifted to the right of the horizontal axis with an increase in the porosity of the root direction region but unchanged porosity of the crown direction region.

#### 3.3.3. Interval Distribution of Equivalent Stress

The interval distribution results of equivalent stress values between T3PC and T3/5 at the BII (Figure 6c) showed that the proportion of nodes whose S < 1.6 MPa was significantly lower in T3/5 than in T3PC. In addition, the proportion of nodes whose stress value was within 1.6 MPa ≤ S < 40 MPa increased significantly in T3/5 compared with T3PC. Moreover, the proportion of nodes whose S ≥ 40 MPa was comparable between T3/5 and T3PC, with no significant increase. These results indicate that GP implants can alleviate stress shielding in cancellous bone without increasing the stress concentration area in the alveolar ridge margin.

The stress nephogram of each GP implant model is shown in Figure 9, and the interval distribution of equivalent stress is shown in Table 7 and Figure 10. The results were as follows: (1) The proportion of nodes whose S < 1.6 MPa decreased gradually, and the proportion of nodes whose equivalent stress value was in the interval of 1.6 MPa ≤ S < 40 MPa and S ≥ 40 MPa (under oblique load) increased gradually with an increase in the porosity of the root direction region but unchanged porosity of the crown direction region. (2) A similar variation trend was observed with an increase in the porosity of the crown direction region but unchanged porosity of the root direction region. (3) Compared with the T0 model, the distribution area of GP implants within the optimal stress interval (1.6 MPa ≤ S < 40 MPa) increased by a maximum of 2.01 times, which is greater than that of most HP implants. (4) Under a vertical load, no stress values of the area at the BII were located within the over-limit stress interval (S ≥ 40 MPa) in GP implants. (5) Under an oblique load, compared with the T0 model, the proportion of nodes whose S ≥ 40 MPa increased by a maximum of 4.56 times in GP implants, which is much smaller than the minimum increase of 11.98 times in HP implants.

In summary, from the perspective of the interval distribution of equivalent stress, the T3/5 and T4/5 models showed the best performance among the GP implants, followed by the T1/5, T2/4, and T2/5 models. However, under a 118.2 N oblique and 120 N vertical load, the proportion of nodes whose stress value was within 1.6 MPa ≤ S < 40 MPa in the T4/5 model was only 1.01 times and 0.996 times, respectively, that of the T3/5 model, which means both models were basically equal. Further, under a 118.2 N oblique load, the proportion of nodes with S ≥ 40 MPa in the T4/5 model was 1.23 times that of the T3/5 model, suggesting that it was significantly higher. Therefore, considering the stress distribution pattern at the BII, T3/5 was considered the best GP implant design.

## 4. Discussion

In terms of the mechanical compatibility of dental implants, the more uniform the stress distribution at the BII, the more favorable the overall mechanical properties [32,33,34]. In this case, bone remodeling causes little change in bone density in the peri-implant area, suggesting better dental implant survival [35]. HP implants have the disadvantages of low mechanical strength, single structural properties, and potential pathological bone stress. Given the above constraints, we established GP implants with high-strength structural design and different pore parameters to explore a porous implant design model with optimal mechanical compatibility.

### 4.1. Stress Distribution Regularity and Ideal Stress Range

When two materials with different elastic moduli are exposed to stress together, the one with a higher elastic modulus bears more stress, while the part with a smaller elastic modulus has almost no stress transfer, which is called the stress-shielding effect. Therefore, in dental implants, the stress generated by mastication is mainly concentrated in the cortical bone, especially in the cortical bone margin, while the stress value in the cancellous bone of the non-apex region is generally very low [36,37,38,39]. In addition, a local high-stress zone is located in the cancellous bone in the apical region, which is mainly related to the load direction acting on the implant. The stress distribution of each implant model in this study conformed to the above rules, indicating that the modeling quality in this study is up to standard.

Harold M. Frost summarized and proposed the relationship between bone remodeling and stress at the beginning of this century [40]. (1) When bone stress is less than 1–2 MPa (strain is less than 50–100 με), bone disuse atrophy occurs, bone growth and remodeling stop, and bone mass decreases. (2) When bone stress ranges from 2 to 40 MPa (strain is 100–2000 με), bone remodeling occurs, and bone mass increases to varying degrees under the stimulation of appropriate stress. (3) When bone stress ranges from 40 to 60 MPa (strain is 2000–3000 με), the bone tissue is in a slightly overloaded state. (a) When the bone cell activity is high and the bone tissue adaptive modification ability is strong, it is manifested as bone structure remodeling and the deposition of new bone. (b) When the bone tissue activity is poor and the adaptability is insufficient, bone microfracture occurs, causing pathological bone resorption. (4) When bone stress is greater than 60 MPa (strain is greater than 3000 με), it is close to the yield point of the cortical bone, and the stress overload causes pathological bone injury. (5) When bone stress reaches 120 MPa (strain is greater than 25,000 με), the fracture strength of the bone has been reached. Geng et al. [41] also showed that in alveolar bone, an equivalent force of 1.6 MPa can prevent the disuse atrophy of bone tissue caused by low stress, which is consistent with Frost’s statement.

Therefore, equivalent stress was divided into three intervals in the present study: S < 1.6 MPa, 1.6 ≤ S < 40 MPa, and S ≥ 40 MPa, representing insufficient mechanical stress stimulation, the optimal stress range, and stress overload, respectively.

### 4.2. Effect of the GP Implant on Stress Distribution

In terms of adjusting the stress distribution, implants with a low elastic modulus facilitate load transfer and show better stress jumping along the BII [42], which can effectively alleviate stress shielding during bone remodeling. The relationship between the elastic modulus *E* of porous materials and porosity *p* is shown in Formula (2) [43,44].
(2)$$\;E=E_{0}{\left[1-p\right]}^{n}$$
where *E*_0_ is the elastic modulus of the solid porous material, *p* is the porosity, and *n* is a constant that depends on the microstructure. The elastic modulus of the porous implant decreases with an increase in porosity.

The results of equivalent stress interval distribution showed that the proportion of nodes with a stress value within the optimal stress range in each GP implant was significantly higher compared with that in the control group. These results indicated that the stress shielding in the cancellous bone was significantly alleviated, which is consistent with the expected hypothesis. In terms of stress overload, under a 120 N vertical load, the area of stress overload in each GP implant was always 0, which is an ideal state. Although the growth rate of the stress overload area at the BII of GP implants was significantly lower than that of HP implants (11.98–30.22) under a 118.2 N oblique load, it still had a growth rate of 2.57–4.56 compared with the control group. However, this does not imply that the design of the GP implant is a failure.

Under a 118.2 N oblique load, the maximum equivalent stress value in the cortical bone in HP implants varied from 200 MPa to 400 MPa (Figure 5a), far exceeding the upper limit of the optimal stress range and even exceeding the fracture strength of the cortical bone by 120 MPa. However, the maximum equivalent stress value in cortical bone in GP implants was 47.54 MPa, with a range (x_max_–x_min_) of 4.51 MPa and a standard deviation of 1.268 MPa (Table 6 and Figure 7b). The stress overload part was all in the range of 40–60 MPa, that is, a slightly overloaded range in Harold M. Frost’s bone remodeling theory [40], and the fluctuation range of 47.54 ± 2.49 MPa was far from 60 MPa, which is closer to the yield point of bone. Except for patients with significantly reduced bone activity (such as radiotherapy patients, diabetic patients, heavy glucocorticoid users, etc.), those who meet the indications for dental implantation can theoretically complete the adaptive remodeling of alveolar bone. In addition, the stimulation of mild overload stress is conducive to bone structure remodeling and new bone deposition over existing alveolar bone, especially for people with high osteocyte activity and strong adaptive reconstruction ability of the bone tissue [45,46]. This is beneficial for young patients who have suffered horizontal or vertical resorption of the alveolar process due to the prolonged absence of teeth.

Therefore, GP implants may be applied when the alveolar bone conditions are not extreme and have several advantages [17,25,47,48]: (1) the benefits of a mild overload state outweigh the disadvantages in the early and middle stages of osseointegration; (2) the excessive local stress of marginal cortical bone will gradually decrease below the safe level with the progress of osseointegration, which ensures the long-term fatigue resistance of porous implants; and (3) in terms of long-term effects, the new bone and pore structure are mutually anchored, which can significantly reduce the vertical bone resorption of the alveolar bone compared with the non-porous implants in the control group.

### 4.3. The Clinical Implications of the Findings

In current clinical practice, the connection between titanium implants and alveolar bone tissue forms a rigid bond. This often leads to stress concentration in the cortical bone and stress shielding in the cancellous bone. In such cases, the load and displacement of the implant follow a linear pattern, which is not conducive to absorbing and distributing stress [1,35]. This study shows that the axial gradient porous structure of the implant can alleviate the stress shielding in cancellous bone without increasing the stress concentration in the cortical bone margin, which solves the problem of poor mechanical adaptability of the implant due to the large elastic modulus ratio of the implant to bone tissue. It can effectively improve the resistance of the implant to adverse stress and extend the clinical service life and comfort of the implant, showing potential for many clinical implications.

### 4.4. Study Limitations

Nonetheless, this study has some drawbacks. In terms of research design, to improve the accuracy and efficiency of the finite element numerical solution, the material properties of the cortical bone and cancellous bone were set as isotropic in this study. However, in practice, the alveolar bone is an anisotropic material, and the porosity of the cancellous bone gradually increases from the margin to the center [49]. Therefore, in addition to finite element research, materials science and biomaterial research on GP implants are also warranted for further verification.

Regarding the application of GP implants, two key considerations emerge. Firstly, there exist individual variances in alveolar bone strength within the population. Hence, it becomes imperative to tailor the pore parameters of GP implants to align with the specific bone conditions of each patient [50]. Secondly, as osseointegration advances, any mechanical complications with GP implants can result in greater bone damage during implant removal [51].

## 5. Conclusions

When dental implants are subjected to masticatory pressure, the stress concentration is prominent on the neck and apex areas of the BII.The GP implants (as shown in Figure A2) meet the ideal strength requirements, making them suitable for implant applications.GP implants can alleviate the stress shielding in cancellous bone without increasing the stress concentration in the cortical bone margin, thus achieving the purpose of optimizing the stress distribution pattern at the BII.Under the premise of ensuring the quality of the preparation process, T3/5 (for which, as shown in Figure A2, the porosity of the crown direction region is 55% and the porosity of the root direction region is 75%) is the best GP implant design.

## Figures and Tables

**Figure 1 jfb-14-00557-f001:**
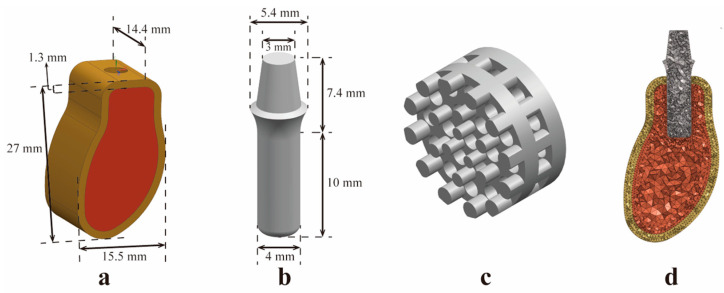
Geometric and grid models. (**a**) Dimensions of 3D mandibular model. (**b**) Dimensions of the 3D implant models. (**c**) Details of porous structure design. (**d**) Details of the finite element grids: the mesh model is dominated by a 0.4 mm C3D10M grid, and a 0.2 mm C3D10M grid is applied to the contact interface between the alveolar bone and the implant for local encryption.

**Figure 2 jfb-14-00557-f002:**
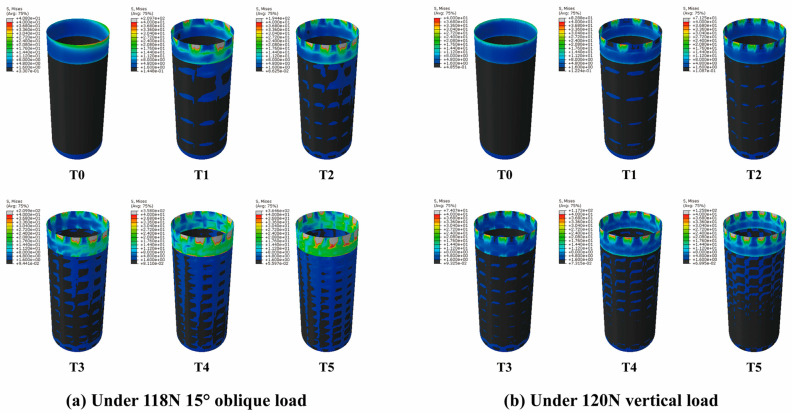
(**a**) The stress nephograms of T0–T5 models under a vertical load of 120 N. (**b**) The stress nephograms of T0–T5 models under an oblique load of 118.2 N. The equivalent stress values in the gray area are all greater than 40 MPa, and the equivalent stress values in the black area are all less than 1.6 MPa. The equivalent stress value gradually increases from the blue to the red area.

**Figure 3 jfb-14-00557-f003:**
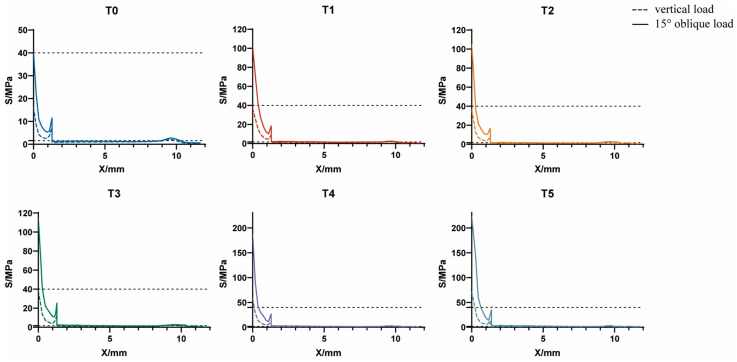
Axial variation trend of equivalent stress in alveolar bone in T0–T5 models.

**Figure 4 jfb-14-00557-f004:**
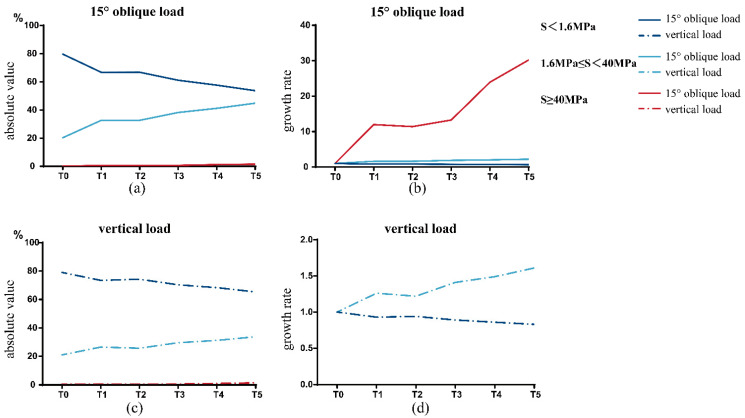
Interval distribution of equivalent stress in T0–T5 models at the BII. (**a**) and (**c**) are the absolute percentage of the area of each equivalent stress interval to the total area of BII. (**b**) and (**d**) are the growth rate of the area occupied by each equivalent stress interval.

**Figure 5 jfb-14-00557-f005:**
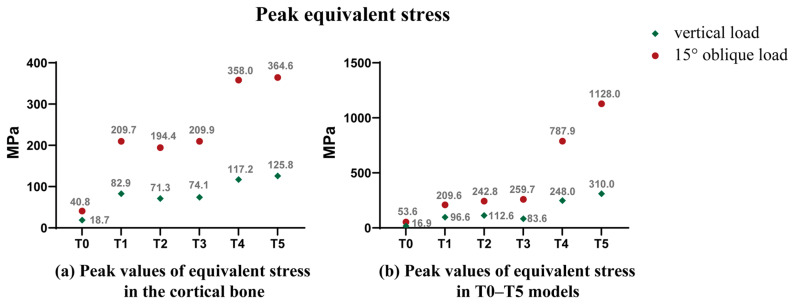
(**a**) Peak values of equivalent stress in the cortical bone; (**b**) Peak values of equivalent stress in T0–T5 models.

**Figure 6 jfb-14-00557-f006:**
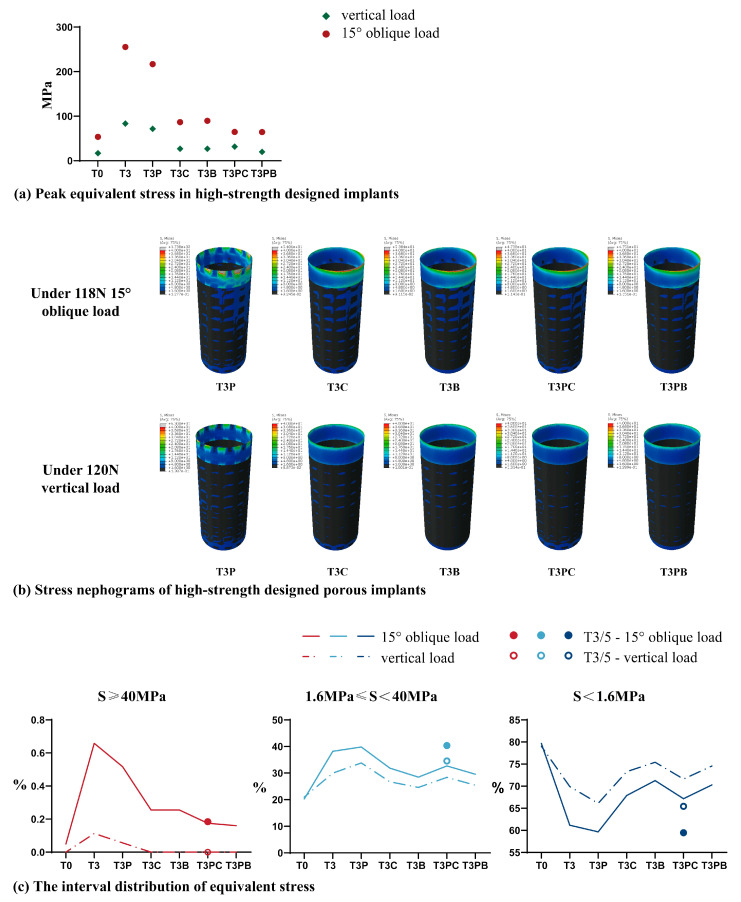
Stress distribution of high-strength designed porous implants. (**a**) Peak equivalent stress in implants. (**b**) Stress nephograms of high-strength designed porous implants; the significance of each index in these nephograms is the same as that in Figure 2. (**c**) The interval distribution of equivalent stress.

**Figure 7 jfb-14-00557-f007:**
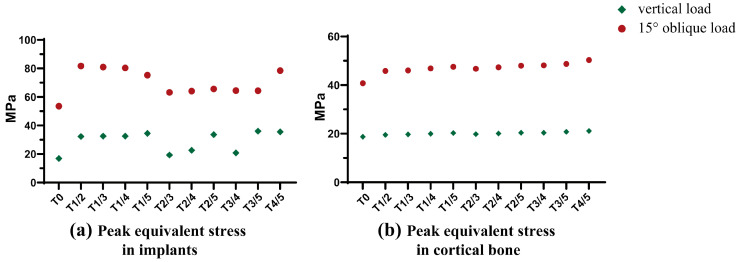
(**a**) Peak equivalent stress in implants; (**b**) Peak equivalent stress in cortical.

**Figure 8 jfb-14-00557-f008:**
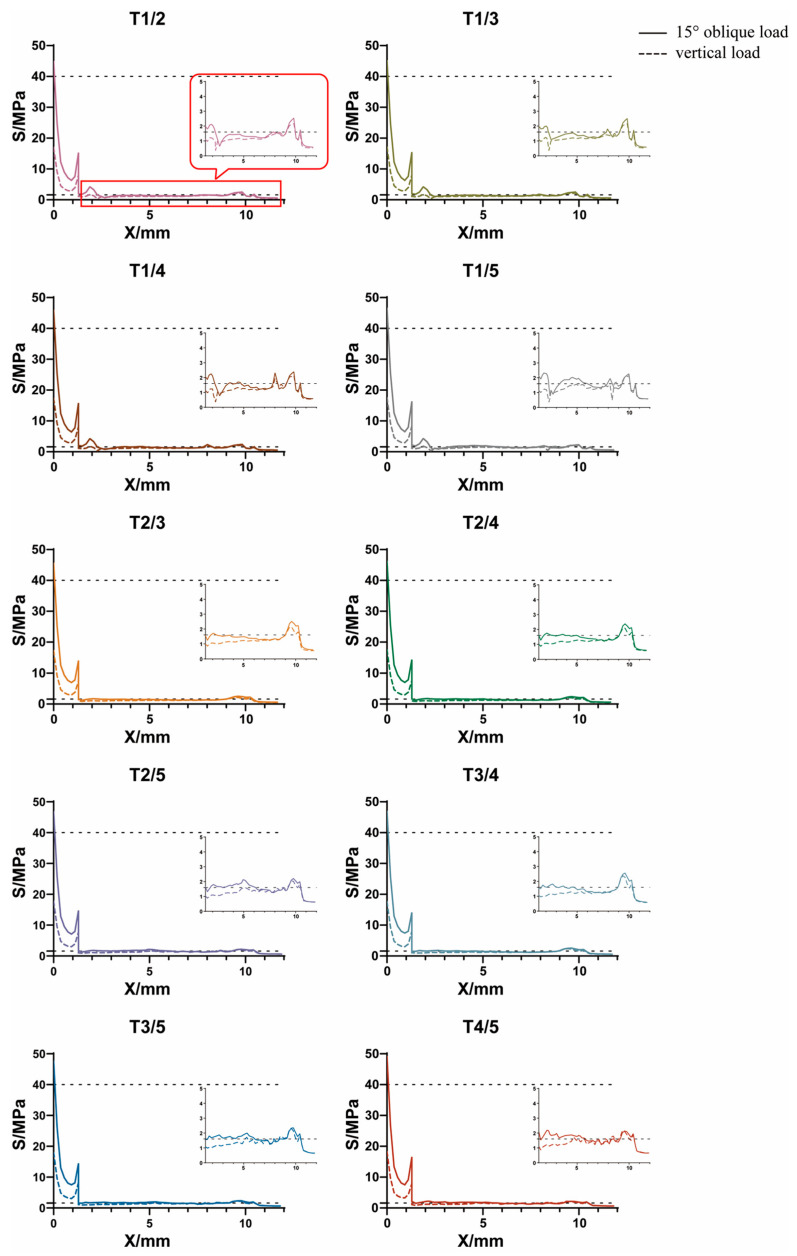
Axial variation trend of equivalent stress values of the alveolar bone in each GP implant model.

**Figure 9 jfb-14-00557-f009:**
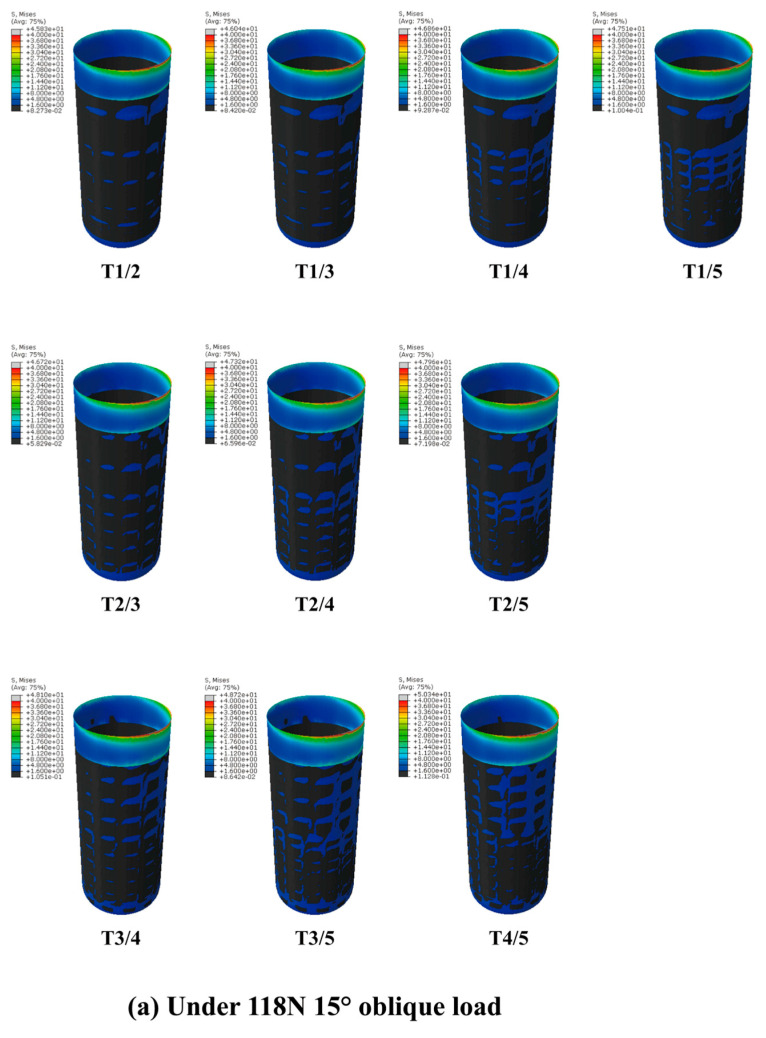
(**a**) The stress nephograms of GP implant models under a vertical load of 120 N. (**b**) The stress nephograms of GP implant models under an oblique load of 118.2 N. The significance of each index in these nephograms is the same as that in Figure 2.

**Figure 10 jfb-14-00557-f010:**
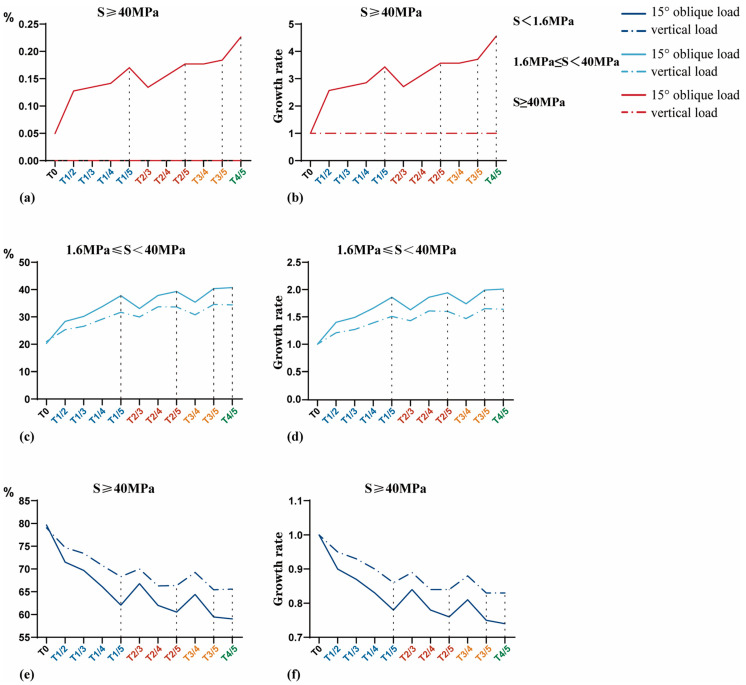
Interval distribution of equivalent stress values of GP implant models at the BII. (**a**,**c**,**e**) are the absolute percentage of the area of each equivalent stress interval to the total area of BII. (**b**,**d**,**e**) are the growth rates of the area occupied by each equivalent stress interval. In the abscissa axis, models with the same porosity in the crown direction region are indicated by the same color.

**Table 1 jfb-14-00557-t001:** Ten sets of gradient porous implant models.

	T1/2	T1/3	T1/4	T1/5	T2/3	T2/4	T2/5	T3/4	T3/5	T4/5
Porosity in the crown direction (%)	35	35	35	35	45	45	45	55	55	65
Porosity in the root direction (%)	45	55	65	75	55	65	75	65	75	75

**Table 2 jfb-14-00557-t002:** Number of elements and nodes for each model.

Components	Number of Elements	Number of Nodes	Components	Number of Elements	Number of Nodes
Cancellous bone	140,067	29,474	T1/2	112,942	25,931
Cortical bone	102,561	24,164	T1/3	113,687	26,382
T0	38,715	8742	T1/4	119,426	28,469
T1	112,898	27,734	T1/5	116,513	28,689
T2	131,591	34,511	T2/3	118,881	28,600
T3	117,549	32,813	T2/4	122,488	30,220
T4	114,341	34,422	T2/5	120,861	30,616
T5	181,394	57,613	T3/4	116,900	29,490
			T3/5	114,689	29,786
			T4/5	120,602	31,660

**Table 4 jfb-14-00557-t004:** Interval distribution of equivalent stress in T0–T5 models at the BII.

Model	S < 1.6 MPa		1.6 MPa ≤ S < 40 MPa		S ≥ 40 MPa	
	Oblique Load (%)	Vertical Load (%)	Oblique Load (%)	Vertical Load (%)	Oblique Load (%)	Vertical Load (%)
T0	79.65	79.03	20.30	20.97	0.05	0
T1	66.74	73.42	32.67	26.48	0.59	0.11
T2	66.78	74.25	32.65	25.68	0.57	0.07
T3	61.13	70.32	38.22	29.58	0.66	0.10
T4	57.63	68.32	41.18	31.1	1.19	0.49
T5	53.71	65.28	44.80	33.77	1.50	0.94

**Table 5 jfb-14-00557-t005:** Interval distribution of equivalent stress values of high-strength designed porous implants at the BII.

Model	S < 1.6 MPa		1.6 MPa ≤ S < 40 MPa		S ≥ 40 MPa	
	Oblique Load (%)	Vertical Load (%)	Oblique Load (%)	Vertical Load (%)	Oblique Load (%)	Vertical Load (%)
T3P	59.65	66.12	39.83	33.83	0.52	0.06
T3C	67.89	73.29	31.86	26.71	0.25	0
T3B	71.26	75.41	28.48	24.59	0.25	0
T3PC	67.15	71.62	32.67	28.39	0.18	0
T3PB	70.29	74.56	29.55	25.44	0.16	0

**Table 6 jfb-14-00557-t006:** Maximum equivalent stress values of GP and HP implants.

Load Type	Implant Type	S_Max_ (MPa)	Mean	Range	Standard Deviation
118.2 N oblique load	GP implants	45.83, 46.04, 46.86, 47.51, 46.72, 47.32, 47.96, 48.10, 48.72, 50.34	47.54	4.51	1.268
	HP implants	209.7, 194.4, 209.9, 358.0, 364.6	267.30	170.20	76.968
120 N vertical load	GP implants	19.53, 19.74, 19.97, 20.29, 19.81, 20.09, 20.36, 20.38, 20.77, 21.12	20.21	1.59	0.460
	HP implants	82.88, 71.25, 74.07, 117.2, 125.8	94.24	54.55	22.749

Bilateral 95% reference range: $\bar{X}\pm 1.96S$.

**Table 7 jfb-14-00557-t007:** Interval distribution of equivalent stress values of GP implants at the BII.

Model	S < 1.6 MPa		1.6 MPa ≤ S < 40 MPa		S ≥ 40 MPa	
	Oblique Load (%)	Vertical Load (%)	Oblique Load (%)	Vertical Load (%)	Oblique Load (%)	Vertical Load (%)
T0	79.65	79.03	20.30	20.97	0.05	0
T1/2	71.50	74.70	28.37	25.30	0.13	0
T1/3	69.68	73.44	30.18	26.56%	0.13	0
T1/4	66.11	70.80	33.75	29.20	0.14	0
T1/5	62.06	68.31	37.77	31.69	0.17	0
T2/3	66.79	70.01	33.08	29.99	0.13	0
T2/4	62.00	66.30	37.84	33.70	0.16	0
T2/5	60.53	66.36	39.29	33.64	0.18	0
T3/4	64.42	69.23	35.40	30.77	0.18	0
T3/5	59.47	65.43	40.34	34.57	0.18	0
T4/5	59.05	65.60	40.72	34.40	0.23	0

## Data Availability

The data presented in this study are available upon request from the corresponding author.

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
