# Peer review of "Biomechanical Analysis of Axial Gradient Porous Dental Implants: A Finite Element Analysis"

_jfb, 2023, doi:10.3390/jfb14120557_

Round 1

Reviewer 1 Report

Comments and Suggestions for Authors

Dear Authors,

The manuscript titled: “ Biomechanical analysis of axial gradient porous dental implants: A finite element analysis“ has interesting aim,  however I have a few suggestions/questions to clarify parts of the manuscript. Please look at my notes and revise the manuscript accordingly.

The Title of the manuscript well conveys with the major concern of the study.

The Abstract is well written.

The Introduction section well sets up the main topic and introduce the development of the manuscript. However, the aim and null hypothesis are not expressed. The null hypothesis statement must precisely identify the variables assessed through statistical analysis. Please add these statements in the last part of the Introduction section.

Other parts of manuscript are well written and discussed.

My only advice is to rewrite conclusion part a little bit easier to understand, it is a little confusing for readers in this way.

Reviewer 2 Report

Comments and Suggestions for Authors

Comments on the Quality of English Language

Please read the article carefully and improve the English language

Reviewer 3 Report

Comments and Suggestions for Authors

The scientific article titled "Biomechanical analysis of axial gradient porous dental implants: A finite element analysis" presents an interesting study on the biomechanical effects of axial gradient porous dental implants. However, there are several aspects o be addressed before the manuscript can be recommended for publishing:

§  Abstract Clarity: The abstract provides an overview of the study, but it could be clearer. The language and structure are somewhat technical, making it challenging for non-experts to grasp the significance of the findings. The abstract could be more reader-friendly.

§  Problem Statement: The article introduces the concept of porous dental implants as a potential solution to alleviate bone stress shielding. Still, it doesn't clearly highlight the specific problem or gap in existing knowledge that this research addresses.

§  Methodology: While the finite element analysis is a robust method for studying biomechanical phenomena, the article lacks a detailed explanation of the modeling and analysis techniques used. Readers may appreciate more information on the assumptions made during the modeling process.

§  Results Interpretation: The article presents findings related to the axial gradient porous implant, but the discussion of these results could be more extensive. A more thorough interpretation of the biomechanical implications and potential clinical relevance would enhance the article's quality.

§  Conclusion: The conclusion should provide a concise summary of the key findings and their practical implications. It should also address the limitations of the study and suggest directions for future research.

§  Citations and References: Ensure that all claims, especially those related to prior research, are properly cited and referenced. This strengthens the article's credibility and demonstrates a robust literature review. The literature review could be strengthened by discussing the very recent relevant articles dealing with Functionally graded materials FGM and porous FG. Further, the use of finite element analysis in dental implant research over the past five years appears to be insufficiently acknowledged in the citations. As a result, I recommend that the authors include a more comprehensive coverage of these recent studies within the Introduction of the manuscript. For instance, the work available could be incorporated to strengthen the introductory framework

Finite element analysis of the effect of porosity on biomechanical behaviour of functionally graded dental implant”. Proceedings of the Institution of Mechanical Engineers, Part E: Journal of Process Mechanical Engineering, 09544089231197857 doi.org/10.1177/09544089231197857

In summary, the article shows promise in exploring the biomechanical effects of axial gradient porous dental implants. However, the manuscript is recommended for publishing provided the above comments are adequately addressed.

Comments on the Quality of English Language

The language and writing style of the article may be more appealing to readers. It is critical to strike a balance between technical information and accessibility for a wider scientific readership.

Author Response

Dear Reviewer,

We feel great thanks for your professional review work on our article. As you are concerned, there are several problems that need to be addressed. According to your nice suggestions, we have made extensive corrections to our previous draft.

In this revised vision, we have addressed your concerns and edited the English language. The Abstract, Introduction, Methods, Discussion, and Conclusion have been further modified, the detailed corrections are listed below, and the revision was marked in red fronts in the manuscript. We hope that these revisions successfully address your concerns and requirements and that this manuscript will be accepted. Looking forward to hearing from you soon.

Best wishes.

Zhang Chunyu on behalf of all authors

Reviewer 4 Report

Comments and Suggestions for Authors

See the attachment 

Comments on the Quality of English Language

Minor English improvement is required 

Round 2

Reviewer 2 Report

Comments and Suggestions for Authors

I thank you the authors for the corrections made, and also for the clarifications. This is an important work to enrich previous work in the field of biomechanics.

I wish you success and continuation

Author Response

Dear reviewer,

      I would like to sincerely thank you for your recognition and encouragement of my thesis work, which is a great inspiration for me. And, thank you for your valuable comments on my paper during the review process, which will play an important role in promoting our subsequent research work.

Reviewer 3 Report

Comments and Suggestions for Authors

The revision meet the reviewer's requirements, so the revised manuscript can be accepted for publication.

Author Response

(The authors gave the same response as above.)
